# IRES Trans-Acting Factors, Key Actors of the Stress Response

**DOI:** 10.3390/ijms20040924

**Published:** 2019-02-20

**Authors:** Anne-Claire Godet, Florian David, Fransky Hantelys, Florence Tatin, Eric Lacazette, Barbara Garmy-Susini, Anne-Catherine Prats

**Affiliations:** UMR 1048-I2MC, Inserm, Université de Toulouse, UT3, 31432 Toulouse cedex 4, France; anne-claire.godet@inserm.fr (A.-C.G.); florian.david@inserm.fr (F.D.); fransky.hantelys@gmail.com (F.H.); florence.tatin@inserm.fr (F.T.); eric.lacazette@inserm.fr (E.L.); barbara.garmy-susini@inserm.fr (B.G.-S.)

**Keywords:** gene regulation, translation, mRNA, IRES, ITAF, hnRNP, chaperone, stress, nucleocytoplasmic translocation, ribosome, lncRNA, translation initiation factor, stress granules, therapeutic targets

## Abstract

The cellular stress response corresponds to the molecular changes that a cell undergoes in response to various environmental stimuli. It induces drastic changes in the regulation of gene expression at transcriptional and posttranscriptional levels. Actually, translation is strongly affected with a blockade of the classical cap-dependent mechanism, whereas alternative mechanisms are activated to support the translation of specific mRNAs. A major mechanism involved in stress-activated translation is the internal ribosome entry site (IRES)-driven initiation. IRESs, first discovered in viral mRNAs, are present in cellular mRNAs coding for master regulators of cell responses, whose expression must be tightly controlled. IRESs allow the translation of these mRNAs in response to different stresses, including DNA damage, amino-acid starvation, hypoxia or endoplasmic reticulum stress, as well as to physiological stimuli such as cell differentiation or synapse network formation. Most IRESs are regulated by IRES trans-acting factor (ITAFs), exerting their action by at least nine different mechanisms. This review presents the history of viral and cellular IRES discovery as well as an update of the reported ITAFs regulating cellular mRNA translation and of their different mechanisms of action. The impact of ITAFs on the coordinated expression of mRNA families and consequences in cell physiology and diseases are also highlighted.

## 1. Introduction

Internal ribosome entry sites (IRESs) are translation regulatory elements of mRNAs that were first discovered in viruses. Until the late 1980s, it was thought that eukaryote mRNAs could not be translated by internal ribosome entry and that the only mechanism was the cap-dependent process involving the recruitment of the small ribosome subunit at the mRNA 5’ end, followed by ribosome scanning [1,2]. This dogma has been proven incorrect with the discovery, in 1988, of RNA structural elements present in the mRNA 5’ untranslated regions (5’UTR) of two picornaviruses, poliovirus (PV) and encephalomyocarditis virus (EMCV), able to mediate cap-independent translation through internal ribosome entry [3,4]. Picornavirus mRNAs are uncapped, with start codons located several hundred nucleotides downstream from mRNA 5’ end, rendering an improbable translation initiation at these AUG codons by the 5’ end-dependent scanning mechanism. Furthermore, these viruses express a protease that cleaves the initiation factor eIF4G, a component of the cap-binding complex, which blocks the cap-dependent initiation process and results in a translational shutdown in the infected cell. The translational machinery is, thus, fully available for the viral mRNA and occurs in a cap-independent manner. Ribosomes are recruited by internal entry via RNA structural domains called “ribosome landing pad” or “internal ribosome entry site” (IRES) that were shown later to exist in different virus families including *Picornaviridae, Flaviviridae, Dicistroviridae* and *Retroviridae* [3,4,5,6,7,8,9,10,11,12], as well as in cellular mRNAs [13,14].

## 2. The Viral IRESs 

Four major IRES classes have been defined in viruses that differ by their mode of ribosome recruitment and secondary/tertiary structure. Type I and II IRESs, found in picornaviruses, are long (400–500 nt long) and present a strong conservation of primary and secondary sequences within each of the two classes [7,15,16]. Their main mechanistic difference is that the type I IRESs (including PV IRES) recruit the ribosome far upstream from the authentic initiation codon; thus, ribosome internal entry is followed by ribosome scanning to reach the start codon. In contrast, the type II IRESs (including EMCV IRES) recruit the ribosome directly onto the initiation codon that is located just downstream from the IRES and do not necessitate ribosome scanning to promote translation initiation. The third important class, whose prototype is hepatitis C (HCV) IRES, concerns the *Flaviviridae* (including HCV) and HCV-like picornaviruses [9,17]. This third class of IRESs is characterized by the presence of a pseudoknot upstream from the AUG codon and by the requirement of the first 30 nt of the coding sequence [18,19,20]. These IRESs are shorter than the Type I and II IRESs and recruit the ribosome directly onto the AUG codon. Intergenic region (IGR) IRESs constitute a fourth class of IRESs, originally identified in cricket paralysis virus (CrPV) [12,15]. IGR IRESs are conserved among members of the *Dicistroviridae* family, whose mRNA is naturally bicistronic. IGR IRESs function in the absence of any start codon. For CrPV, translation starts at a GCU triplet. Moreover, these IRESs can form 80S ribosomes without the initiator Met-tRNA [12]. *Retroviridae* IRESs, whose mRNAs are capped, resemble cellular mRNA IRESs (see below).

## 3. The Cellular IRESs

Soon after the finding of the two first IRESs in picornaviruses in 1988, two host trans-acting factors, La autoantigen and pyrimidine tract binding protein (PTB), were identified as IRES-binding factors required for internal initiation of translation [21,22,23,24]. This suggested that the internal initiation process might also concern cellular mRNAs, although these mRNAs are capped. Actually, the first IRES mediated by the 5’ leader of a cellular mRNA was described in 1991 in the immunoglobulin heavy-chain binding protein (BiP) mRNA [13]. What could be the usefulness for a capped mRNA to contain an IRES? The first hypothesis was that IRESs could allow cellular mRNA translation when the cap-dependent process is blocked, which was known to occur during mitosis (G2-M phase) and in stress conditions such heat shock or viral infection [25,26,27]. Favoring this hypothesis, the Bip messenger codes for a chaperone involved in the unfolded protein response occurring during endoplasmic reticulum (ER) stress, and its synthesis was detected in spite of the translation blockade generated by poliovirus infection [13]. Although this first cellular mRNA IRES was indicative of a major role of IRES-dependent translation in the stress response, the physiological relevance of IRESs in the translation of cellular mRNAs was questioned during many years because these mRNAs are capped in contrast to the picornavirus mRNAs. Nevertheless, it quickly became clear that the BiP mRNA was not a unique case: IRESs were found in a series of other cellular mRNAs, including transcription factors such as the homeobox (Hox) gene *antennapedia*, the proto-oncogene c-*myc*, angiogenic growth factors, such as fibroblast growth factors (FGFs) and vascular endothelial growth factors (VEGFs), as well as many genes coding for the master regulators of cell responses [14,28,29,30,31,32,33,34]. Interestingly, it was shown that these capped IRES-containing mRNAs may be translated either by the cap-dependent or by the IRES-dependent mechanism, according to the conditions: a switch from cap- to IRES-dependent mechanism during hypoxia has been described for VEGFA and HIF1α IRESs in breast cancer resulting from the overexpression of eIF4G and of 4E-BP, a protein that sequesters the cap-binding protein eIF4E [35]. This switch has also been reported for the p53 mRNA in conditions of oncogene-induced senescence and for the VEGFC IRES under hypoxia [29,36]. As viral IRESs, cellular IRESs imply RNA secondary structures that are conserved in mammals [29,30,37,38]. However, in contrast to viral IRESs, they cannot be classified as they do not exhibit sequence or secondary structure similarities. 

The function of cellular IRESs clearly appears in the case of mRNAs that are naturally bi- or multicistronic. A few dozen mRNAs have been reported that express two or more proteins, in most cases via IRES elements [39]. Two specific cases have been reported: mRNAs with distinct open reading frames (ORF) separated by an intergenic region and mRNAs with overlapping open reading frames leading to the synthesis of proteins with a common C-terminal portion but differing in their N-terminal part. The latter case can be illustrated by the examples of angiogenic growth factors FGF2 and VEGFA, whose mRNAs are subjected to alternative initiations of translation [40,41]. The FGF2 mRNA uses four CUGs and one AUG to express five FGF2 isoforms, which have different localizations and functions [40,42]. Translation from the upstream CUG is cap-dependent whereas all the other start codons are used by an IRES-dependent mechanism. With regards to the VEGFA, its mRNA contains two IRESs driving translation from two initiation codons CUG and AUG, leading again to isoforms with different intracellular localizations [28,41].

Only few bi- or multicistronic mRNAs with intergenic regions have been discovered [39]. The c-myc mRNA is tricistronic when transcribed from the upstream alternative promoter P0 in human [31,43]. It contains three distinct ORFs separated by intergenic regions containing IRESs. These two IRESs drive the translation of the middle ORF and MYCHEX1 and of the downstream ORF that codes for the c-myc1/c-myc2 proteins (initiated at two alternative start codons) [43]. The MDP6 antigen is coded by an ORF located in the 3’ region of the myotrophin mRNA, and its synthesis is controlled by an IRES sensitive to interferon-α [44]. Thus, the translation of the two ORFs of this bicistronic mRNA are differently regulated. A bicistronic mRNA has been also described in rat brain, coding for the two subunits of a glutamate-binding protein complex in rat brain synaptic membranes, PRO1 and PRO2 [45]. These two polypeptides are synthetized from the bicistronic mRNA, implying an IRES in the intergenic sequence, which allows their coordinate expression. The same observation has been made for the two G-coupled fatty acid receptors GPR40 and GPR41 [46].

The physiological relevance of IRESs present in monocistronic mRNAs clearly appeared with the discovery of the X-linked inhibitor of apoptosis (XIAP) mRNA IRES [47]. This IRES was shown to be induced in apoptotic conditions. This observation was also made for other IRESs of mRNAs coding for factors involved in apoptosis, including the apoptotic peptidase activating factor 1 (APAF1), c-*myc* and p53 [48,49,50,51]. These findings definitely highlighted the crucial role of IRES-dependent translation for cellular mRNAs. Actually, during apoptosis, the cap-dependent translation process is blocked as it is after the picornavirus infection due to the cleavage of eIF4G [52]. XIAP and APAF1 have opposite functions during apoptosis; thus, their relative level due to differential IRES activation is essential for the life/death decision of the cell in the progression of the apoptosis pathway [52].

Thereafter, the IRES physiological function was evidenced in several reports. First, an important tissue specificity of cellular IRES activities was observed in contrast to the picornavirus IRES activity. This was revealed with the FGF2 IRES in transgenic mice: the IRES was inactive in almost all adult organs, except for the brain and testis, where the activity was very strong, much stronger than in cultured cells [53]. Further investigation demonstrated that the FGF2 IRES is a key of FGF2 translational induction during spermatogenesis and during the formation of synaptic networks between neurons [54,55]. In contrast, the activity of the FGF1 IRES, another member of the FGF family, is strong in skeletal muscle and involved in the control of FGF1 expression during myoblast differentiation and muscle regeneration [56].

In addition to their roles in specific adult organs, IRESs are important in the control of gene expression during development. The FGF2 and c-*myc* IRESs were as active as the strong EMCV (encephalomyocarditis virus) IRES in mice embryo in contrast to what was observed in adult [53,57]. The early discovery of an IRES in the mRNA of the homeobox Hox gene *Antennapedia* in Drosophila also argued such a hypothesis, whereas the proof of concept, which definitely demonstrated the key role of IRESs in development, was provided 23 years later by Maria Barna and her collaborators who identified IRESs in four HoxA mRNAs [32,58]. These authors have shown that these IRESs are conserved in evolution and demonstrated that they are essential for mouse development by generating the first targeted mouse knockout of a cellular IRES [58]. Moreover, the presence of IRESs in cellular mRNAs was investigated in a high throughput study, which uncovered thousands of sequences, allowing cap-independent translation, and which showed that 10% of the mRNAs harbour cap-independent sequences [59]. Although the methodology used in this report may include false positives, it indicates that IRES-dependent translation may concern a large number of cellular mRNAs and, thus, constitutes, under the control of ITAFs, a pivotal mechanism in the control of gene expression.

The dysfunction of IRES-dependent translation has been related to various pathologies. A single mutation in the c-*myc* IRES is responsible for c-myc overexpression in multiple myeloma [60]. Also, single mutations in the connexin 32 and VEGFA IRESs have revealed the essential role of IRESs in two severe neurodegenerative diseases, Charcot-Marie-Tooth disease and amyotrophic lateral sclerosis, respectively [61,62]. More recently, an aberrant increase of IRES-dependent translation of key cancer gene mRNAs has been reported in cancer cells, including the major angiogenic factors FGF1, FGF2 and VEGFA, as well as c-*myc* and insulin growth factor-like receptor (IGF1R) [63]. This study revealed that the mechanism of IRES activation results from p53 tumor suppressor inactivation: p53 represses the expression of the rRNA methyl-transferase fibrillarin, which modifies the rRNA methylation pattern and generates “cancer ribosomes” that will be preferentially recruited by IRES-containing mRNAs.

These different studies of IRES-related pathophysiological functions demonstrate the key role of IRES-dependent translation, revealing the coexistence of cap-dependent and -independent translation for capped mRNAs containing IRESs. They also provide exciting therapeutic opportunities to specifically regulate the expression of IRES-containing genes (often involved in the control of cell proliferation, cell death, angiogenesis, etc.) at the translational level by targeting different ITAFs.

## 4. IRES-Dependent Translation, a Pivotal Mechanism in the Stress Response

Physiological and environmental stresses induce drastic changes in the regulation of gene expression, which permits cell adaptation and survival, or in contrast, triggers programmed cell death. It was long believed that such changes mostly occur at the transcriptional level. For example, hypoxia, which is one of the major physiological stresses during development, generates the stabilization of the hypoxia-induced factor (HIF) which induces the transcription of a series of target genes. However, this strong transcriptional response is not the only way to modify gene expression during stress. Several posttranscriptional mechanisms were shown to participate in the hypoxic response, among which translational control plays a key role. 

Above all, global translation is blocked during stress to save energy, as translation is estimated to consume up to 50% of cellular energy [52]. This translation blockade is observed in most stress conditions including hypoxia, nutrient limitation, temperature changes, ultraviolet irradiation, endoplasmic reticulum stress, oxidative stress, viral infection, etc. The two main locks of this blockade are the translation initiation factors eIF4E and eIF2α [64]. The first way of inhibiting translation results from the mechanistic target of rapamycin (mTOR) kinase inactivation, which induces hypophosphorylation of 4E-binding proteins (4E-BPs). When dephosphorylated, 4E-BP sequesters the cap-binding protein eIF4E, generating a blockade of cap-dependent translation. The second way of translation inhibition by stress is due to eIF2α phosphorylation, which blocks the exchange of GDP to GTP in the eIF2 complex and prevents the assembly of the ternary complex eIF2-GTP-tRNA_i_^Met^ required for the binding of the initiator Met-tRNA_i_^Met^ to the 40S ribosomal subunit. There are four known stress-responsive eIF2α kinases able to impact global translation: haem-regulated inhibitor kinase (HRI), protein kinase RNA (PKR), PKR-like endoplasmic reticulum kinase (PERK) and general control non-derepressible-2 (GCN2). These kinases are activated by different stresses that induce a common pathway of translation blockade [52,64]. The eIF2 pathway subtlety is that it also induces the selective translation of transcripts, mostly coding for master regulators of the cell responses, including transcription factors, growth factors, etc. Such a selective translation occurs by two main initiation mechanisms: small upstream open reading frame (uORF)-regulated initiation and IRES-driven initiation. The best documented example of translation initiation regulated by uORFs is the yeast transcriptional activator GCN4 [65]. The GCN4 mRNA contains four uORFs upstream from the GCN4 ORF. When the ternary complex eIF2-GTP-tRNA_i_^Met^ is abundant, uORFs are translated, which prevents the translation of the GCN4 ORF. In contrast, if the level of ternary complex is low, under amino-acid starvation, scanning ribosomes fail to initiate at the uORFs and translate the GCN4 ORF. This mechanism has also been described for the mammalian transcription factor ATF4 [64].

IRES-dependent translation, the focus of the present review, is the other main mechanism of selective translation upon eIF2α phosphorylation. Although eIF2α is, in principle, required for both cap-dependent and independent translation, IRES-dependent translation is selectively increased in the condition of phosphorylated eIF2α. This was first observed for the IRES of the Arg/Lys transporter cat-1, as well as for several viral IRESs [66]. Interestingly, the activation of cat-1 IRES observed in response to amino-acid starvation, ER stress and double stranded RNA requires eIF2α phosphorylation by GCN2, PERK and PKR, respectively. This suggests that the cat-1 IRES can function efficiently when the level of ternary complex eIF2-GTP-tRNA_i_^Met^ is low. This was also observed for BiP, XIAP and other stress-responsive transcript IRESs [67,68]. Two models have been proposed for this intriguing observation: i) ribosome recruitment and formation of the initiation complex utilizes initiation factor 5B (eIF5B) that delivers the tRNA directly into the P site of the ribosome to form a translation-competent initiation complex [68,69] and ii) IRES-dependent translation is increased following the transcriptional induction of 4E-BP by GCN2 and its downstream transcription factor, activating transcription factor 4 (ATF4): In conditions of limiting ternary complex eIF2-GTP-tRNA_i_^Met^, a stronger blockade of cap-dependent translation by 4E-BP results in increased IRES-dependent translation [67].

The upregulation of IRES-dependent translation by stress has an impact in various pathologies. For instance, hypoxia appears in the center of solid tumors exceeding a volume of two cubic millimeters which are not any more irrigated by blood vessels. As the major angiogenic and lymphangiogenic growth factors of the FGF and VEGF families possess IRESs in their mRNAs, these growth factors are translationally induced as their IRESs are sensitive to hypoxia [29,35,70,71]. This results in tumoral angiogenesis and lymphangiogenesis, two processes that promote tumor cell invasion and metastasis dissemination. Hypoxic stress also occurs in cardiovascular diseases such as lower limb ischemia and ischemic heart disease. In these pathologies, cells are subjected to hypoxia due to artery occlusion in the ischemic leg or in an infarcted myocard. In particular, chronic heart failure is a public health issue. IRES-dependent translation plays a major role during ischemia: a very recent mid-scale study shows that, unexpectedly, the expression of most (lymph)angiogenic factors is not induced at the transcriptome level but at the translatome level in hypoxic cardiomyocytes [72]. The same study indicates that the IRESs of (lymph)angiogenic factors mRNAs, FGF1, FGF2, VEFGA, VEGFC and VEGFD are activated in early hypoxia, while non angiogenic IRESs such as EMCV or c-*myc* IRES are activated in late hypoxia. Furthermore, the FGF1 IRES is also activated in an ischemic heart in vivo in a mouse model of an infarcted myocard [73]. The IRES-dependent translation in an ischemic myocard thus allows a rapid angiogenic response that contributes to cardiomyocyte survival. These data enlighten the strong pathophysiological impact of IRES-dependent translation to stimulate tumoral and non-tumoral (lymph)angiogenesis in response to hypoxia (Figure 1).

## 5. IRES Trans-Acting Factors, Key Regulators of Cellular IRESs

Most IRESs, and in particular cellular IRESs, require IRES trans-acting factors (ITAFs) to function in addition to several canonical translation initiation factors. Around fifty proteins have been described for their ability to specifically regulate cellular IRESs, while a single long noncoding RNA (lncRNA), TP53-regulated modulator of p27 (TRMP), is also able to regulate IRES-dependent translation (Table 1) [74].

A near-exhaustive bibliographic analysis of ITAFs controlling cellular IRESs has been performed here, revealing several classes of ITAFs. The largest class is composed of nuclear proteins able to shuttle from the nucleus to the cytoplasm to control the IRES-dependent translation. This class contains many heterogeneous nuclear ribonucleoproteins (hnRNPs) but also other proteins such as nucleolin, HuR or p54^nrb^ (Table 1). A second ITAF class is composed of cytoplasmic proteins. Most of them are translation machinery-associated proteins with ribosomal and ribosome-associated proteins as well as translation initiation or elongation factors. However, other cytoplasmic or membrane-associated proteins have an ITAF function, whereas they have no reported interaction (at the moment) with the translation machinery except for being an ITAF. For instance, upstream of the N-ras (Unr) is a cytoplasmic cold shock protein which is also associated to the endoplasmic reticulum; hepsin is a plasmic membrane-associated protein able to control the Unr mRNA IRES, while vasohibin 1 (VASH-1) is a mostly cytoplasmic and secreted protein known for its antiangiogenic and stress resistance features before being identified as an ITAF (Table 1). A third ITAF class contains, at the moment, a single member, the 834 nucleotide long non-coding (lnc) RNA TRMP, inhibitor of the p27kip IRES [74]. Another ncRNA, miR-122, has been described but only for a viral IRES. This small liver-specific RNA binds to the HCV 5’UTR and competes with PCBP2 binding, which blocks translation and increases the viral RNA engagement in replication versus translation [75]. The discovery of an ITAF function exhibited by a lncRNA is very recent; thus, one can expect that TRMP is probably not the only lncRNA to regulate the IRES-dependent translation, as many lncRNAs could serve as assembly platforms for regulatory proteins. Interestingly, TRMP is an inhibitor of the p27kip IRES and is a direct transcriptional target of p53 itself, regulated at the IRES-dependent level by sixteen reported ITAFs (Table 1).

It has been often reported that viral IRESs harbor specific secondary or tertiary structures with common domains while it is difficult to identify any structural conservation between different cellular IRESs [15,74]. Despite this difference, many reported ITAFs seem to control the IRES-dependent initiation of translation for both cellular and viral IRESs. A well-documented example is PTB (also known as hnRNPI), first described as an ITAF of the EMCV IRES [22]: This protein is able to modulate the translation of a dozen reported virus IRESs as well as at least fourteen cellular IRESs (Table 1) [152]. The IRES binding sites of this protein have been extensively studied. Six PTB binding sites have been mapped on the EMCV IRES, corresponding to unpaired oligopyrimidine tracts [22,171]. Two of them are crucial for the IRES activity, located on a stem-loop structure 400 nt upstream from the AUG and just upstream from the AUG codon [22,172]. The binding of these two PTB molecules might be required to stabilize the IRES conformation. The PTB binding to cellular IRESs has also been studied and can be illustrated by the example of the APAF1 IRES [151]. PTB and its neuronal variant nPTB have two binding sites, the major one being located in an exposed loop in the 3’ part of the APAF1 IRES. The other ITAF controlling the APAF1 IRES, unr, binds to a purine-rich loop in an upstream domain of the IRES [151].

ITAFs able to drive ribosome recruitment on both viral and cellular IRESs are also found among class II cytoplasmic ITAFs: A nice example is provided by the ribosomal protein rps25 (eS25), required for *Discitroviridae*, *Flaviviridae*, *Picornaviridae* and *Retroviridae* IRES activities, as well as for at least ten cellular IRESs (Table 1) [165,173,174]. Unr is able to regulate viral IRESs such as poliovirus and human rhinovirus IRESs, as well as at least five cellular IRESs [15]. These observations indicate that the same ITAFs can control viral and cellular IRESs, suggesting that either ITAFs act via a similar mechanism for the two IRES types or a given ITAF can act by different mechanisms. An important difference between viral and cellular IRESs is the stronger tissue-specificity of cellular IRESs [53]. One can hypothesize that cellular IRESs require specific ITAFs expressed only in certain cell types or tissues and/or that they are more susceptible to negative ITAFs that would silence these IRESs in specific tissues. Specific ITAFs could regulate groups of mRNAs in a coordinated manner, thus defining regulons [58]. 

While several IRESs can be regulated by a same ITAF, a given IRES can be regulated by several ITAFs, which may be positive or negative regulators. As shown in Table 1, we have listed thirteen cellular ITAFs able to inhibit IRES-dependent translation. Furthermore, ten of them have the double role of IRES activator or inhibitor, depending on the IRES. Among the best-documented IRESs regulated by several ITAFs are the p53 mRNA IRESs [175]. Two p53 IRESs have been described, controlling the expression of either the full-length p53 (FL-p53) or of a p53 isoform devoid of N-terminal domain, ΔN-p53. These two IRESs are induced by genotoxic or cytotoxic stress. In basal non-stressed conditions, the IRES activity is inhibited by two negative ITAFs, nucleolin and programmed cell death protein 4 (PDCD4), whereas two other ITAFs, translational control protein 80 (TCP80) and RNA helicase A (RHA), are bound to the RNA but with an inadequate interaction that cannot activate the IRES. Following stress, the interaction of TCP80 and RHA is increased and several other positive ITAFs including ribosomal protein RPL26 (uL24) and hnRNPQ bind to the IRES, facilitating secondary structure unwinding and enhancing IRES activity. The ΔN-p53 mRNA IRES is activated during stress by several other ITAFs including PTB, death-associated protein 5 (DAP5), PTB-associated splicing factor (PSF) and Annexin A2 [175]. In addition, proteins bound to the 3’UTR of the FL-p53 mRNA also influence the IRES activity: The protein Quaking has an inhibitory effect on the IRES activity while HuR binds to the 3’UTR during stress, displaces Quaking and activates translation. It is likely that many IRESs, as well as p53 IRESs, are regulated by a protein complex rather than by a single ITAF. The composition of this complex, called the IRESome, varies among IRESs and is probably a means to regulate the IRES activity specifically. The presence of different partners in the complex may also help us to understand why a given ITAF can be either negative or positive depending on the IRES, as shown for at least nine ITAFs (Table 1). 

The cryogenic electron microscopy (cryo-EM) has allowed a real technological advance to study the ribosome assembly on the IRES [176]. Only viral IRESs have been studied at this time. Two first studies of the HCV IRES bound to 40S or 80S ribosomes at low resolution cryo-EM (between 15 and 30 Å) have revealed that the HCV IRES RNA induces a change in the 40S ribosome conformation that promotes translation initiation, while the bound 80S ribosome generates structural rearrangements in the two IRES pseudoknots [177,178]. A few years later, high resolution cryo-EM studies of ribosome interactions (3.7 to 3.9 Å) with the CrPV, HCV and classical swine fever virus (CSFV) IRESs have provided detailed information about the molecular interactions within the ribosome-IRES complexes [179,180,181,182]. Cryo-EM has revealed that the CrPV IRES pseudoknot structure interacts with the decoding center of the 40S ribosome and necessitates a translocation before it can accept an aminoacyl tRNA to start translation. Also, an atomic model of the ribosome-bound HCV IRES RNA has been proposed, showing the precise interactions of the IRES with different ribosomal proteins and with the 18S rRNA. IRES interaction with the ribosomal protein eS27 prevents the binding of eIF3 to a ribosome, which decreases the cap-dependent translation [180,181]. As eIF3 directly binds to the HCV IRES and activates the IRES activity, it appears that the role of eIF3 in IRES-dependent translation differs from its canonical role of interaction with the cap-binding factor eIF4F [180,181].

A cryo-EM analysis remains to be done for the translation initiation complexes with cellular IRESs. The data for viral IRESs indicate that this powerful approach will allow the deciphering of the molecular interactions of ITAFs with IRESs and ribosomes to better understand their modes of action. 

## 6. Multifunctional ITAFs: How are They Assigned to the Translational Function?

ITAFs have often other functions in addition to their role in IRES-dependent translation. Most of them have been first discovered for playing roles in alternative splicing (hnRNPs), ribosome biogenesis (nucleolin, TCP80, RHA), mRNA stability (HuR), transcription (p54^nrb^, hnRNPK, −M, RHA, SMAR1), etc. The question of how they are assigned to their translational function remains to be investigated. However, several reports provide some answers. The first one is the intracellular localization. Numerous multifunctional ITAFs are mainly nuclear proteins that can translocate in the cytoplasm. A well-documented example is hnRNPA1 [37,94,98,183]: This protein is relocalized to the cytoplasm in stress conditions, resulting in IRES negative or positive regulations. HnRNPA1 activates FGF2 and sterol regulatory element-binding protein 1 (SREBP-1) IRESs while it inhibits APAF1 and XIAP IRESs. Such a relocalization has been reported for other ITAFs, including PTB and poly r(C) binding protein 1 (PCBP1; also known as hnRNPE) that act in concert to activate the Bcl-2-associated athanogene 1 (BAG1) IRES in response to chemotoxic stress [110]. Also, nucleolin is translocated from nucleolus to cytoplasm to activate the VEGFD IRES in response to heat shock [30].

ITAF activity is also regulated by various posttranslational modifications. This was first demonstrated for RNA-binding motif protein 4 (RBM4), an ITAF described, at the moment, only for viral IRESs [184]. Following arsenite exposure, RBM4 is phosphorylated, which accompanies its cytoplasmic relocalization and targets stress granules. When phosphorylated, RBM4 both inhibits cap-dependent translation and activates IRES-dependent translation. With regards to cellular IRESs, hnRNPA1 constitutes a well-documented example for the role of posttranslational modifications: its binding to c-myc and cyclin D IRESs is regulated by Akt phosphorylation [95]. Furthermore, hnRNPA1 dimethylation on its glycine-arginine-rich (GAR) motif by the type II arginine transferase PRMT5 is required for the activation of cyclin D1, c-myc, HIF1α and estrogen receptor α (ER-α) IRESs [185]. Another ITAF described more recently, the tumor suppressor PDCD4, is phosphorylated by protein kinase S6K1 or Akt and subsequently degraded via the ubiquitin ligase β-TCRP [140,142,143,186]. PDCD4 inhibits cap-dependent translation, while it is a negative or positive ITAF depending on the IRES: It is a repressor of p53, BcL-XL and XIAP and an activator of INR and IGF1R IRESs [140,141,143,186]. Multiple posttranslational modifications have been described for hnRNPK, a protein overexpressed in many cancers [187]. This multifunctional protein is subjected to phosphorylation, methylation, ubiquitination and sumoylation while it interacts with diverse groups of molecular partners involved in transcription, chromatin remodeling, RNA processing, translation and signal transduction [188]. hnRNPK sumoylation on a lysine residue promotes its ITAF function and results in the activation of the c-*myc* IRES in Burkitt’s lymphoma cells [187].

## 7. ITAFs Use Different Mechanisms of Action to Control IRES-Dependent Translation

We have seen above that ITAF activities are regulated by different parameters including nucleocytoplasmic shuttling, posttranslational modifications and the interaction with diverse partners. The question is by which mechanisms are ITAFs able to activate or inhibit IRES-dependent translation? As described below, nine ITAF mechanisms have been documented (Figure 2).

### 7.1. Chaperones

The first mechanism to be described is a role of chaperone for PTB (and its neuronal form nPTB) and Unr [151]. These two proteins are required to activate the APAF1 IRES and act by altering the secondary structure of the IRES. According to the report by Mitchell et al, Unr first binds to two stem loops identified in the IRES, generating a conformational change that renders it accessible to the nPTB or PTB binding sites [151]. Then a second conformational change occurs, providing the correct conformation for the 40S ribosome subunit binding. The cooperation of two or more ITAFs in IRES activation through an RNA conformational change has been described for other IRESs: The BAG1 IRES is also controlled by a couple of ITAFs, PTB and PCBP1 [189]. Again, there is a successive binding of the two ITAFs, with, first, PCBP1 that opens the RNA, allowing PTB binding and subsequent 40S recruitment. In these studies, PTB appears as an essential part of the preinitiation complex.

The different hnRNPs described as ITAFs are likely to act by the chaperone mechanism, often in conjunction with other mechanisms. By example, the chaperone mechanism can be associated to competition and nucleocytoplasmic translocation mechanisms, as described below.

### 7.2. Competitive Binding

The interplay between different ITAFs can be competitive rather than cooperative: This is the case of Annexin A2, PSF and PTB [76]. Annexin A2 and PSF would act as chaperones or by stabilizing the preinitiation complex as shown for PTB. These three ITAFs are all activators of the second IRES present in the p53 mRNA between the FL-p53 and ΔN-p53 AUG codons [175]. However, they compete for IRES binding as they share overlapping binding sites. Annexin A2 binding is calcium-dependent whereas PSF binding is not. The authors propose that the accumulation of more calcium ions in the cytoplasm during ER stress would promote Annexin A2 binding to activate the IRES activity, whereas PSF and PTB would play a role in other stress conditions or physiological stimuli. Actually, it has been proposed that PTB regulates the differential expression of p53 isoforms during the cell cycle and in response to DNA damage [148]. Competitive binding has also been reported for couples of ITAFs harboring opposite activities. HuR and hnRNPC compete for their binding to the IGF1R IRES, which is silenced by HuR and activated by hnRNPC [106]. The lncRNA TRMP inhibits the p27^kip^ IRES activity by competing with the IRES for PTB binding and thus preventing IRES activation mediated by PTB [74].

### 7.3. Nucleocytoplasmic Translocation

The role of nucleocytoplasmic translocation of many ITAFs in IRES activation (Table 1) does not answer the question of ITAF nuclear or cytoplasmic binding. Actually, the ITAF can be translocated to the cytoplasm upon stress and then can bind to the IRES-containing mRNA, or it can bind to the IRES in the nucleus and then be translocated with the IRES-containing mRNA as a ribonucleoprotein. In such a case, the ITAF can also play a role in the nuclear retention of the IRES-containing mRNA in the absence of stress [82]. Clearly, the regulation of APAF1 IRES by the successive binding of Unr and PTB suggests that PTB binds to this IRES in the cytoplasm because Unr is cytoplasmic. In contrast, a set of arguments indicates that certain ITAFs bind to the IRES in the nucleus. In particular, a nuclear event is required for a IRES-dependent translation controlled by certain IRESs: this has been shown for c-*myc* and FGF1 IRESs by demonstrating that these IRESs are not able to drive translation when the cells are transfected with a bicistronic in vitro-transcribed mRNA, while the same IRESs are active upon DNA transfection, implying mRNA transcription in the nucleus [50,117]. It is different for viral HRV and EMCV IRESs which exhibit a similar activity following RNA or DNA transfection, showing that the nuclear event is not required for all IRESs. 

### 7.4. Promoter-Dependent Recruitment

A second argument favoring the existence of the nuclear recruitment of ITAFs onto the IRES is brought by the discovery of a mechanism of coupling between translation and transcription for the FGF1 IRES [56]. The activity of FGF1 IRES is promoter-dependent, a mechanism explained by ITAF recruitment onto the promoter that facilitates the recruitment on the mRNA. These two ITAFs, hnRNPM and p54^nrb^, are able to enhance both transcription and translation: first, they activate the FGF1 promoter and then the FGF1 IRES-dependent translation. The dependence to promoter does not change the mechanism of action for ribosome recruitment. However, one can expect that such ITAFs act as chaperones to stabilize the IRES conformation very early in the mRNA maturation process. Although it has not been demonstrated that these ITAFs can bind on the nascent mRNA in a co-transcriptional manner, it is a plausible hypothesis, as co-transcriptional binding has been shown for several hnRNPs during the alternative splicing process [117,190].

### 7.5. Interaction with Translation Initiation Factors or with 4E-BP

Additional ITAF mechanisms of action have been discovered that strictly occur in the cytoplasm during the translation initiation process. Several ITAFs act by inhibiting translation initiation factors. RBM4 was shown to interact with the initiation factor 4A (eIF4A) in response to arsenite treatment, which promotes the association of eIF4A with the IRES-containing mRNA [184]. By this way, RBM4 simultaneously activates IRES- and inhibits cap-dependent translation, probably by stabilizing the eIF4A interaction with the IRES which facilitates the unwinding by eIF4A of the IRES domain necessary for ribosome recruitment. Indeed, it has been reported that the IRES-dependent recruitment of ribosome 43S requires RNA unwinding [191]. Interestingly, this is concomitant with RBM4 targeting stress granules. RBM4 has not yet been shown to regulate any cellular IRES; however, its interaction with eIF4A and with eIF2β has been shown for DAP5, an ITAF of the eIF4G family. DAP5 regulates several cellular IRESs of the genes involved in apoptosis as well as its own IRES [83]. Another ITAF acting via eIF4A interaction is PDCD4, whose interaction with eIF4A was demonstrated by crystal structure and mutation analysis, while it also interacts directly with the IRES [142]. However, in contrast to RBM4, PDCD4 has been described as a negative ITAF and acts by blocking the helicase activity of eIF4A [140]. PDCD4 would thus prevent the unwinding of the IRES ribosome recruitment domain by eIF4A. This ITAF controls at least five IRESs in p53, INR, IGF1R, BcL-XL and XIAP mRNAs, most of them involved in apoptosis, also revealing the relevance of the eIF4A binding mechanism for cellular IRESs (Table 1). 

PTEN-induced putative kinase-1 (PINK1), involved in Parkinson’s disease, activates the HIF1α mRNA translation during hypoxia by acting on 4E-BP1 [35,168]. It has been shown that PINK1 stimulates the switch of the 4E-BP hyperphosphorylated γ form (inactive form) to the hypophosphorylated α form (active form) that sequesters eIF4E and inhibits the cap-dependent translation, while it activates IRES-dependent translation by increasing the availability of eIF4G for IRES-dependent translation. PINK1 acts on 4E-BP1 as well as on 4E-BP2, the predominant 4E-BP protein in the brain. The activator effect of PINK1 has been shown only for EMCV IRES; however, the decrease of HIF1α mRNA translation in PINK^−/−^ mouse strongly suggests that PINK1 is also an activator of the HIF1α IRES [168]. The authors do not rule out that PINK1 could affect the activity of other translation factors such as S6K, eIF4E, eIF4G, eEF2 or eIF2α.

### 7.6. ITAF Role of eIFs, eEFs and 4E-BP

The role of canonical translation initiation in the activity of viral IRESs has been intensively addressed in the mid-1990s when IRES-mediated initiation was reconstituted in vitro, showing that EMCV IRES activity requires the same canonical factors as the cap-dependent initiation: eIF2, -3, -4A, -4B and -4F [192]. eIF4F, composed of the cap-binding protein eIF4E as well as of eIF4A and eIF4G, binds to the IRES in a cap-independent manner, and its activity is attributed to its eIF4A and -4G. It was demonstrated that eIF4A and the central domain of eIF4G are sufficient to mediate the ribosome 43S recruitment [191]. In the case of other viral IRESs, HCV and CSFV, the initiation factor eIF3 is able to bind to specific domains of these IRESs [193]. The process of ribosome recruitment onto these IRESs is exceptional because it does not involve the canonical factors eIF4A, -4B or -4F, while eIF3 is absolutely required for ribosome 80S formation on these IRESs. 

In the 2000s, the specific requirement of several eIFs was also demonstrated for cellular IRES-mediated translation initiation. During apoptosis, the three members of the eIF4G family, eIF4GI, eIF4GII and DAP5, are cleaved by caspases. While the eIF4GII cleavage products are degraded, eIF4GI and DAP5 proteolytic fragments are stable and activate IRESs of pro-death APAF1 and DAP5 mRNAs, a process that will contribute to the fine-tuning of the cell fate [86]. The C-terminal domain of eIF4GI bound to eIF4A is also required for the activities of N- and c-myc IRESs [161]. In *C-elegans*, the eIF4GI ortholog (IFG-1) activates Bip and Bcl-2 IRESs during apoptosis [163]. eIF4GI has as well an ITAF function on the HIF1α and VEGF IRESs in conditions of hypoxia [35]. In that study, it was shown that tumors overexpress both eIF4G and 4E-BP1, which orchestrates a switch of cap-dependent to IRES-dependent translation for cancer key mRNAs resulting from an increased sequestration of eIF-4E. The overexpression of these two proteins promotes tumor growth and angiogenesis. 

Supporting the similarities of viral and cellular IRES-dependent mechanisms, eIF3 is also required for N- and c-myc IRES activities by a direct mechanism that does not necessitate eIF3 binding to eIF4G, resembling the viral mechanism where eIF3 is directly recruited to the mRNA [161]. The XIAP IRES is also activated by direct eIF3 binding via an RNA structural domain located close to the AUG codon [162]. The polyA-binding protein (PABP) binding to eIF3 allows mRNA circularization that promotes ribosome binding. Thus, eIF3 acts as a scaffold for 40S ribosome recruitment.

IRES-dependent translation is also regulated by eIF5B, a factor classically known to be involved in ribosome subunit joining, whose prokaryotic homolog IF2 promotes the binding of initiator tRNA to the prokaryotic 30S ribosomal subunit. This was first shown for the CSFV IRES, whose activity is not affected by eIF2α phosphorylation [194]. eIF5B is able to stabilize the ribosomal binding of initiator tRNA when eIF2 is inactive. This mechanism was also shown for the XIAP IRES: When eIF2α is inactivated, the IRES-dependent translation switches to the eIF5B mode [68]. 

Finally, the elongation factor eEF1A2, whose canonical role is to shuttle aminoacyl-tRNA during translation elongation, has been characterized as an ITAF for the utrophin A IRES, activated during muscle regeneration [160]. Its mechanism of action has not been elucidated, but the hypothesis is that it could interact with a tRNA-like structural element in the utrophin A IRES.

### 7.7. Translocation to Stress Granules and P-Bodies

Stress granules are cytoplasmic foci mainly composed of stalled preinitiation complexes that appear in stress conditions due to eIF2α phosphorylation. Processing bodies (P-bodies) are cytoplasmic structures that assemble during stress after stress granules and are dedicated to the storage and degradation of untranslated mRNA [195]. The current hypothesis is that mRNAs exiting translation would first accumulate in stress granules before being transferred to P-bodies. Interestingly, several ITAFs have been localized in stress granules or P-bodies. PTB is partially localized in stress granules upon infection by cardiovirus Theiler’s murine encephalomyelitis virus (TMEV) [196]. As PTB is an activator of the TMEV IRES, its sequestration in stress granules would make it unavailable for virus genome translation, explaining in part the negative impact of stress granules on viral replication. hnRNPA1 is also a known stress granule-associated protein, giving it a role in cell survival to stress [197]. However, the relation with its ITAF function is not clear. A stress granule component, the ras GTPase SH3 stress granule assembly factor 1 (G3BP1), has been characterized as an inhibitor of foot-and-mouth disease virus (FMDV) IRES [198]. G3BP1 interacts with the FMDV IRES and also interacts with hnRNPA1 and PTB, suggesting that its inhibitory effect could occur by mRNA retention in the stress granule as well as by the sequestration of hnRNPA1 and PTB.

Also, translocation between cytoplasm and processing bodies (P-bodies) has been described for PCBP1 and PCBP2 upon stress conditions [199]. These authors suggest that PCBPs could play a role in shifting rapidly certain untranslated mRNAs into a translationally active state. However, the link between PCBP translocation and IRES-dependent translation has not been elucidated yet. Gemin5, a component of survival of motor neuron (SMN) proteins whose dysfunction is involved in spinal muscular atrophy, interacts with eIF4E and colocalizes with it in P-bodies [200]. Gemin5 is an inhibitor of cap-dependent translation but is also able to bind to FMDV and HCV IRESs and to block their activity, in particular upon serum starvation [201].

### 7.8. Association to Ribosome

Another cytoplasmic mechanism of IRES regulation concerns ribosome-associated proteins. RACK1, a *Drosophila melanogaster* 40S ribosome subunit-associated protein (now considered as a ribosomal protein) was found associated to the HCV IRESome and was shown as an essential determinant of HCV, *Drosophila* C virus (DCV) and CrPV IRES activities [202,203]. RACK is an adaptor protein interacting with a variety of signaling proteins.

Reaper, a potent apoptosis inducer in *Drosophila melanogaster*, inhibits cap-dependent translation by directly binding to the 40S ribosome subunit, while it allows IRES-dependent translation to occur via the Cricket paralysis (CrPV) IRES [204]. Reaper is the first discovered cellular ribosome binding protein to allow a selective translation. Its interaction with a ribosome small subunit in the late stage of initiation might affect the function of eIF5B and inhibit 60S joining or might directly inhibit the recognition of the start codon. Although Reaper has not yet been documented for its effect on cellular IRESs, one can hypothesize that certain cellular IRESs may also be regulated by this mechanism. 

Two other inducers of apoptosis, grim in *Drosophila* and second mitochondrial activator of caspase (smac) in humans, were shown to inhibit translation, suggesting that they could allow the same selective translation as Reaper [205]. Interestingly, Reaper and grim mRNA translation are IRES-regulated [206,207]. In human, the glycogen synthase GYS1 was found associated to polysomes, while its depletion results in the loss of polysomes and affects translation positively or negatively depending on the mRNA [208]. The selectivity of these proteins towards IRES-dependent translation remains, however, unknown.

The SMN component Gemin5, described above for its P-bodies localization, is also a ribosome-associated protein. Indeed, Gemin5 inhibits the FMDV IRES activity by outcompeting PTB via its C-terminal domain, while it inhibits global translation by interacting with the uL3 and uL4 proteins of the 60S ribosomal subunit via its N-terminal domain [209,210]. Recently, Gemin5 was identified as a positive ITAF of the thioredoxin-interacting protein (TXNIP) IRES [89].

### 7.9. Ribosome Inherent Constituent

Finally, it appears that ribosomal proteins can be directly involved in the control of IRES-dependent translation. The ribosome has been viewed during the last decades as an apparatus able to translate the genetic code without having an intrinsic regulatory capacity. However, several reports have shifted the view of ribosome function by revealing the existence of specialized ribosomes with specific features, rendering them able to control gene expression [211]. The first demonstration of a ribosomal protein that is specifically required for IRES-mediated translation initiation, while not necessary for cap-dependent translation, was provided by Landry et al. [212]. These authors have shown that rps25 (eS25) is required for the activation of CrPV and the hepatitis C virus (HCV) IRESs. Additional studies demonstrated that this protein is globally required for viral IRES as well as for cellular IRES activities. Rps25 is an activator of many cellular IRESs including APAF-1, BAG1, c-*myc*, L-*myc*, Myb, p53 and Set7 IRESs (Table 1). Other ribosomal proteins seem to regulate families of messengers, thus defining regulons. It has been documented in a report showing that RPL38 (eL38) is required for ribosome recruitment onto IRESs of the hox gene family, constituted of homeobox genes involved in development [58]. A recent report has definitely demonstrated that heterogeneous ribosomes are able to preferentially translate distinct subpools of mRNAs [166]. This study highlights the role of RPL10A (uL1) in the activation of IGF2, amyloid precursor protein (APP), charged multivesicular body protein 2A (Chmp2A) and Bcl-2 IRESs [166,170]. Such IRES activation would occur by the direct interaction of the ribosomal protein with the IRES, resulting in ribosome recruitment. 

## 8. Discussion

The present update highlights the discovery of about fifty ITAFs able to regulate the IRES-dependent translation of cellular mRNAs. It also recapitulates the extensive work performed on viral IRESs that had a pioneer role in the scientific advances in understanding the mechanisms of IRES-dependent translation. Viral IRESs have paved the way to cellular IRESs. Advanced technologies such as cryo-EM will soon provide information on the interaction of cellular IRESs with ribosomes and reveal the molecular role of ITAFs in such interactions.

The existence of cellular IRESs have long been a controversial topic, as it was more difficult to definitely prove with adequate controls the IRES-dependent translation mechanism of capped mRNA compared to viruses with uncapped mRNAs. Also, it is easier to measure viral replication than a cell biological process to demonstrate the IRES physiological relevance. Due to the presence of the cap-structure at the mRNA 5’end, the only way to demonstrate the presence of an IRES in eukaryotic mRNAs has been to use the bicistronic reporter system. It has been shown that several previously reported IRES structures contain cryptic promoters or splicing acceptor sites, resulting in false positive findings. However, the presence of a cryptic promoter or splicing site is not incompatible with the presence of an IRES. As shown for VEGFA, an alternative promoter has been identified just between the two IRES sequences; nevertheless, the two IRESs are actually operational in the VEGFA mRNA [70]. With regards to XIAP, a cryptic splicing event has been observed but only when the first cistron is the renilla luciferase (LucR), due to cryptic splicing donor sites in the LucR sequence [52]. This problem has been circumvented by using another reporter gene such as beta-galactosidase [52]. Mutating the cryptic splicing sites in the LucR gene is also a good solution [72]. Stringent and carefully controlled experiments such as Northern blots, RT qPCR or first cistron knockdown have been necessary to demonstrate the presence of *bona fide* cellular IRESs [49,51,53,58]. In spite of these difficulties and of several articles claiming the non-existence of cellular IRESs, it has now been evidenced, based on many reports, that the process of IRES-dependent translation not only concerns viral mRNAs but also has a crucial impact in the translational regulation of cellular mRNA expression, in particular in stress conditions. 

The discovery of about fifty ITAFs (the list remains to be completed in the future) able to regulate cellular IRESs indicates that the control of gene expression by the IRES-dependent process is far from marginal. These different ITAFs play a key role in many physiological processes including development, cell differentiation, cell cycle regulation, apoptosis or circadian oscillation. Furthermore, they are pivotal in the cell response to all possible stress conditions (Table 1). Given that ITAFs regulate the expression of families of genes involved in these processes, they have a strong impact in different pathologies. ITAFs are important actors in many cancers but also in cardiovascular diseases such as ischemic heart disease and neurodegenerative diseases including Parkinson’s disease, Alzheimer’s disease or amyotrophic lateral sclerosis. 

The ITAF involvement in many pathologies provides important perspectives to find new targets to regulate the translation of specific genes or gene networks in a therapeutic objective. Actually, this is being investigated for cancer therapeutics. An intensive area of research concerns the search for new molecules that could reactivate p53 expression [175]. The p53 positive ITAFs TCP80 and RHA appear as promising therapeutic targets to increase p53 levels. Their reduced expression seems to be responsible for a decreased p53 response following DNA damage in breast cancer cells expressing the wild type p53: they represent targets whose expression could be stimulated [155]. Also, the knockdown of p53 negative ITAFs may be an attractive strategy. Moreover, ITAFs are of great interest as therapeutic targets because they are able to control the translation of families of mRNAs. For example, an ITAF able to activate the IRESs of several angiogenic growth factors during hypoxia will constitute a target of choice to block angiogenesis. The discovery of ITAFs controlling IRES regulons, thus, opens an avenue to find new therapeutics.

## Figures and Tables

**Figure 1 ijms-20-00924-f001:**
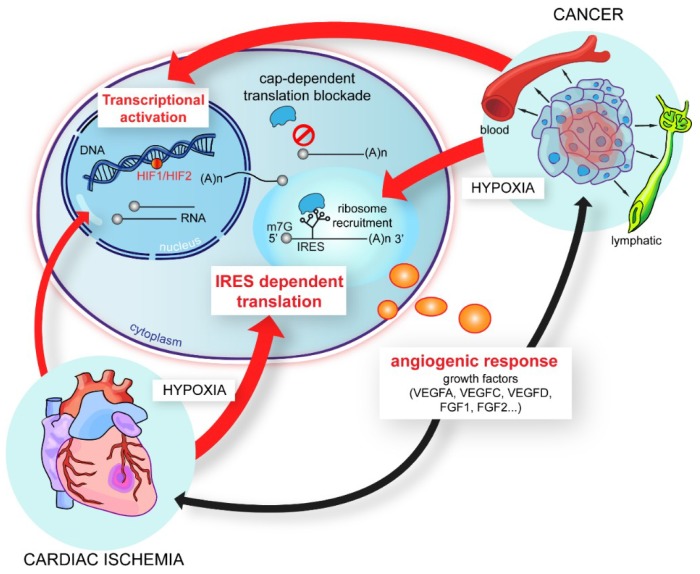
The regulation of the (lymph)angiogenic growth factor expression during hypoxia: (Lymph)angiogenic growth factors are regulated at the transcriptional and/or translational levels during hypoxia. In conditions of tumoral hypoxia, the regulation is both transcriptional and translational through the internal ribosome entry site (IRES)-dependent mechanism, whereas during cardiac ischemia in hypoxic cardiomyocytes, most regulation is translational [72]. The IRESs of (lymph)angiogenic growth factor mRNAs are activated during early hypoxia by an HIF1-independent mechanism.

**Figure 2 ijms-20-00924-f002:**
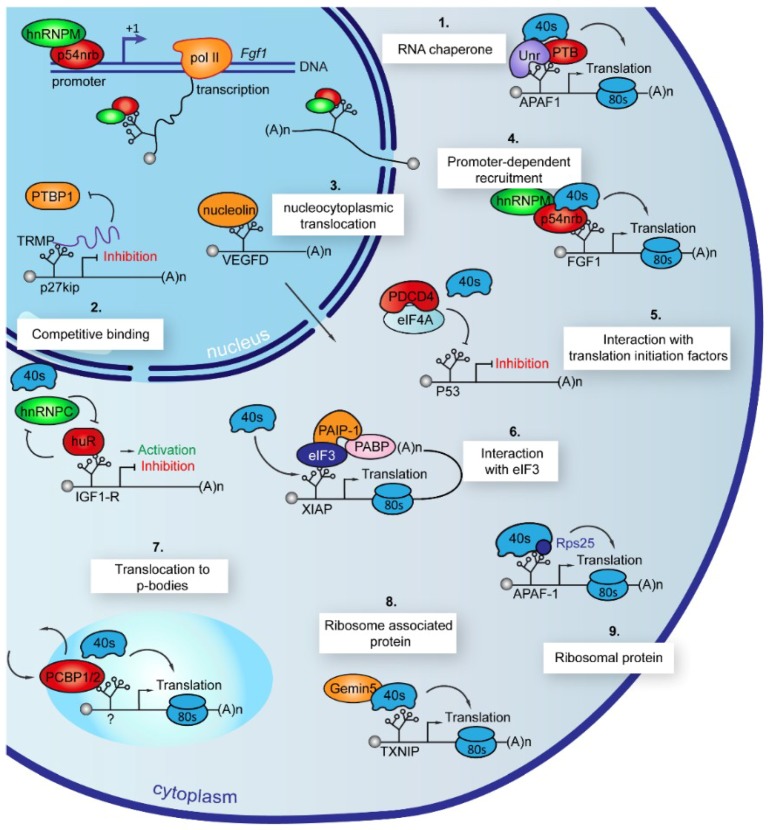
The ITAFs use different mechanisms of action to control IRES-dependent translation. The different reported mechanisms of ITAFs to regulate IRES activities are schematized. For each mechanism, an example is shown with the names of the ITAF and of the IRES. The start point of translation is indicated by an arrow if translation initiation is increased or by a blocked arrow if translation initiation is inhibited. Each mechanism is detailed in the text.

**Table 1 ijms-20-00924-t001:** An update of the reported IRES trans-acting factors (ITAFs) that regulate cellular IRESs. The different reported ITAFs regulating cellular IRESs are indicated. They are dispatched into three classes (see text). For each ITAF, the regulated IRESs, the type of regulation (activator or inhibitor), the described stimuli able to trigger their activity, the roles in cell physiology and diseases as well as the corresponding references are shown.

ITAF	Also Known As	Regulated IRESs	Regulation	Stimulus	Roles in Cell Physiology and Diseases	References
**Class I: ITAFs with nucleocytoplasmic translocation**						
Annexin A2		p53	activator	ER stress	cancer	[76]
CUGBP1	CELF1	SHMT-1, p27kip	inhibitor/activator	UV irradiation	DNA repair, cell proliferation	[77,78]
DAP5	P97, NAT1, eIF4GII	Bcl-2, Bcl-XL, BAX, APAF-1, DAP5, Δ40p53, CDK1, HIAP2, c-*myc*, XIAP	activator	viral infection, apoptosis, ER stress, serum starvation, g-irradiation	cell survival or programmed cell death	[79,80,81,82,83,84,85,86,87,88]
FBP3	FUBP3	TXNIP	activator		Renal cell carcinoma	[89]
FUS		LEF1	activator		Cancer, amyotrophic lateral sclerosis	[90]
GRSF1		c-*myc*, L-*myc*, N-*myc*	activator		cancer	[91]
H-ferritin		SHMT-1	activator	UV irradiation	DNA repair	[77,92]
HDMX		p53	activator	DNA damage	tumour suppression	[93]
hnRNPA1		XIAP, FGF2, Nfil3, SREBP1-a, c-*myc*, BCL-XL, cyclin D1, APAF-1, sst2, ER-α HIF1-α	activator/inhibitor	FGF2, lipid accumulation, ER stress, osmotic shock, UV irradiation	multiple myeloma, circadian oscillation	[37,94,95,96,97,98,99,100,101,102]
hnRNPC	hnRNP C1/C2	p53, IGF1R, unr, c-*myc*, XIAP	activator	DNA damage, transcription inhibition, growth stimulus, cell cycle	inhibition of apoptosis, cancer	[103,104,105,106,107]]
hnRNPD	JKTBP1	NRF	activator	UV irradiation	cell survival	[108,109]
hnRNPE	PCBP, alphaCP	c-*myc*, BAG1	activator	Chemotoxic stress	cell survival, tumorigenesis	[110,111,112]
hnRNPH2		SHMT1	activator	UV irradiation	DNA repair	[77]
hnRNPK		c-myc	activator		myoblast differentiation, proliferation, tumor progression	[111,113]
hnRNPL		Cat-1, p53, LINE-1	activator	Amino-acid deprivation/ DNA damage	transposition inhibition	[114,115,116]
hnRNPM		FGF1	activator	myoblast differentiation	muscle regeneration	[117]
hnRNPQ	NSAP1	p53, rev-erb-a, Period1, AANAT, Bip, FMRP	activator	apoptosis/ heat shock	circadian oscillation/ cell survival/ axonal growth cone collapse/ Fragile X syndroma, autism	[118,119,120,121,122,123]
hnRNPR		AANAT	activator		circadian oscillation	[124]
HuR	ELAV1	IGF1R, caspase-2, BcL-XL, XIAP, p27kip, Thrombomodulin	activator/inhibitor	amino-acid deprivation, IL-1b,	cytoprotection, inhibition of apoptosis, cell proliferation, breast cancer	[106,125,126,127,128,129,130]
La auto antigen		XIAP, Bip, RRBP1	activator/inhibitor	serum starvation, paclitaxel, adriamycin	cell survival, malignancy maintenance, hepatocellular carcinoma	[106,125,126,127,128,129,130,131,132,133]
Mdm2	HDM2	p53, XIAP	activator	DNA damage, ionizing radiation	resistance to radiation-induced apoptosis	[93,134]
NF45		iIAP1, XIAP, NRF, ELG	activator	ER stress	polyploidy, senescence	[135]
nPTB		IR	activator	cell density, insulin	cell proliferation	[136]
nucleolin		p53, VEGFD, LINE-1	activator/inhibitor	heat shock, DNA damage	transposition inhibition	[30,115,137,138]
p54nrb	NONO	c-*myc*, L-*myc*, N-myc, APAF1, FGF1	activator	myoblast differentiation, nucleolar stress, apoptosis	muscle regeneration	[91,117,139]
PDCD4		P53, INR, IGF1R, BcL-XL, XIAP	activator/inhibitor	oxidative stress, absence of DNA damage, S6K2 inactivation, FGF2 pathway inhibition	apoptosis, tumour suppression	[140,141,142,143]
PSF	SFPQ	p53, c-*myc*, L-*myc*, N-*myc*, BAG1, LEF1	activator/inhibitor	nucleolar stress, apoptosis, ER stress	cancer	[76,90,91,139]
PTB	hnRNPI/ PTBP1	p53, p27kip, PFK1, IR, Cat-1, APAF1, HIF1α, IRF2, rev-erb-a, unr, c-*myc*, N-*myc*, BAG1, Bip, ADAR1, TXNIP	activator/inhibitor	DNA damage, hypoxia, ER stress, amino-acid deprivation, cell density, insulin	circadian oscillation, cell cycle arrest, apoptosis	[89,91,107,112,114,136,144,145,146,147,148,149,150,151,152,153,154]
RHA	NDH II	p53	activator	DNA damage	tumour suppression	[155]
SMAR 1		p53	activator/inhibitor	glucose deprivation	cancer (tumor suppressor)	[156]
YB1	YBX1	c-*myc*, L-*myc*, N-*myc*, p16INK4	activator	hypoxia	multiple myeloma, cancer	[91,145,157]
**Class II: Cytoplasmic ITAFs related to translational machinery**						
4E-BP1		VEGFA, HIF1α, INR	activator	hypoxia, low nutrients, low insulin	cancer, Parkinson	[35,158]
APP (AICD)		Δ40p53	activator		Alzheimer disease	[159]
eeF1A2		utrophin A	activator		muscle regeneration	[160]
eIF3		c-*myc*, N-*myc, XIAP*	activator	apoptosis, hypoxia	cancer	[161,162]
eIF4A		c-*myc*, N-*myc*	activator	apoptosis, hypoxia	cancer	[161]
eIF4GI		APAF-1, DAP5, Bcl-2, Bip, c-*myc*, L-*myc*, N-*myc*, VEGFA,	activator	apoptosis, hypoxia	cancer	[35,86,161,163]
eIF5B		*XIAP*	activator	apoptosis, hypoxia	cancer	[68]
eL38	Rpl38	Hox	activator		development	[58]
eS19	Rps19	BAG1, CSDE1, LamB1	activator		erythroid differentiation, Diamond-Blackfan anemia	[164]
eS25	Rps25	APAF-1, BAG1, c-*myc*, L-myc, Myb, p53, Set7	activator	ER stress	multiple myeloma	[101,165,166]
Gemin5		TXNIP	activator/inhibitor	Serum starvation	Spinal muscular atrophy	[89]
Hepsin (also in plasmic membrane)		unr	inhibitor		Cell cycle regulation, Prostate cancer	[167]
PINK1 (also mitochondrial)		HIF1α	activator	hypoxia	Parkinson	[168]
Rack1		IGF1R	activator/inhibitor		Hepatocellular carcinoma	[169]
TCP80	NF90, DRBP76	p53	activator	DNA damage	tumour suppression	[155]
uL1	Rpl10A	IGF2, APP, Chmp2A, Bcl-2	activator		Alzheimer, leukemia, mitochodrial dysfunction	[166,170]
uL24	Rpl26	p53	activator	DNA damage	tumour suppression	[137,138]
uL5	Rpl11	BAG1, CSDE1, LamB1	activator		erythroid differentiation, Diamond-Blackfan anemia	[164]
VASH1 (also secreted and nuclear)	Vasohibin 1	FGF1	activator	hypoxia	ischemic heart disease	[72]
**Class III: ncRNA-constituted ITAFs**						
TRMP		p27kip	inhibitor	induced by p53	inhibition of cell proliferation, tumor suppressor	[74]

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
