# Peer review of "IRES Trans-Acting Factors, Key Actors of the Stress Response"

_ijms, 2019, doi:10.3390/ijms20040924_

Round 1

Reviewer 1 Report

The manuscript by Godet et al reports an update of RNA-binding proteins known as ITAFs (for IRES-transacting factors) regulating IRES activity. Interestingly, the authors propose the existence of different mechanisms by which ITAFs can promote IRES activity. This is a timely review, it is well organised and easy to follow. However, some concerns need to be addressed prior to publication.

1. Some of the ITAFS mechanisms proposed need further clarifications. i) It will be important to better explain how the use of a different promoter determines a distinct mechanism of action of ITAFs. ii) Concerning eIFs, the authors mentioned eIF4A, but nothing is said about eIF3 and other initiation factors. This section is far from being complete. iii) Also, the authors mentioned briefly P-bodies and PCBP1-2. However, the potential translocation of mRNAs and ITAFs to stress granules, a fast cellular response to stress signals, is lacking. iv) The association of ITAFs to the ribosome needs further clarification as well. As it stands, it seems that Reaper is the only protein reported to interact with the ribosome regulating IRES activity.  This is not correct; a few others such as Rack1 and Gemin5 should be included in this point. v) In addition, authors should check the activity of Reaper for IRESs described in flies. vi) A large number of ITAFs are collected in Table 1, yet many of them are not mentioned in the text.

2. The first sentences of the introduction should be rewritten. To clarify how cellular IRESs were discovered, it is critical to report the discovery of prototype viral IRES (picornavirus, HCV, and dicistrovirus), and summarize the critical features of these elements. This is relevant for readers outside the field, as cellular IRES are compared later in the text to picornavirus or to viral IRESs several times in the text. In this regard, it will be important to clarify the type of viral IRES compared to, as not all of them behave the same way. 

3. For the sake of clarity, it is important to describe in the introduction that cellular mRNAs carrying IRES elements are translated using a cap-dependent manner. Authors should then discuss how mRNAs switch from cap-dependent to IRES-dependent translation using specific examples of well validated cellular IRESs. The manuscript lacks information about di- or multi-cistronic cellular mRNAs, and how and when the second cistron could be translated.

4. It is surprising that the sentence in l. 40 does not cover a reference to HCV IRES, reported much earlier than the retrovirus IRESs. Moreover, prior to the work of Hellen et al., 1993, PTB was reported to be an ITAF by Jang et al., 1989, and Luz and Beck, 1991.

5. In l. 37-39 it should be mentioned that picornaviruses induced the cleavage of factors essential for cap-dependent translation (eIF4G). Therefore, not only these RNAs are uncapped, but they can be translated under cap-dependent translation shutdown. It is surprising that this point is lacking in the review, since this strategy was used by P. Sarnow´s team to discover the first cellular IRESs. 

6. The sentence in l. 87 needs to be clarified. The work of Weingarten et al (ref 32) claims, but does not prove, the existence of a large number of IRESs in cellular and viral mRNAs. To put things in perspective, the methodology used in this work and the possibility of including false-positives should be discussed. Moreover, the statement that a short sequence could be responsible for local IRES activity is currently an over-interpretation. Given previous controversies regarding certain published data of IRESs present in cellular RNAs it is important to keep close to the results.

7. Differences in secondary structures between viral and cellular IRESs should be better discussed (l. 209-211), as for most cellular IRESs RNA structure is not known. In addition, viral IRESs also differ in RNA structure. Therefore, the sentence about the role of PTB in IRES activity regardless of RNA structure conservation is misleading. This paragraph is lacking what is known about the binding site of PTB in cellular and viral IRES.  Information about the binding site of the different ITAFs listed in Table 1 is also lacking.

8. The sentence in l. 219-223 regarding ITAFs involved in viral and cellular IRES activity regulation should be modified; it is important to distinguish between binding and mechanism of action. It is surprising that authors first suggest that there are no different mechanisms of action, but then speak about tissue specificity of cellular IRESs. Are there factors conferring tissue specificity? What is the meaning of specific and general ITAFs? May be the mechanism of action is different.

9. The sentence in l. 238 is not complete.

10. Authors should explain what the meaning of eIF4A-binding mechanism is, and its relevance for IRES activity (l. 351).

Author Response

First we wish to sincerely thank the reviewer for his work in reviewing this manuscript. Thanks to these comments that required an important bibliographic work to be answered, we believe this article is much more complete.

The manuscript by Godet et al reports an update of RNA-binding proteins known as ITAFs (for IRES-transacting factors) regulating IRES activity. Interestingly, the authors propose the existence of different mechanisms by which ITAFs can promote IRES activity. This is a timely review, it is well organised and easy to follow. However, some concerns need to be addressed prior to publication.

1.     Some of the ITAFS mechanisms proposed need further clarifications. i) It will be important to better explain how the use of a different promoter determines a distinct mechanism of action of ITAFs.

Indeed, the promoter-dependence does not change the mechanism of action for ribosome recruitment. Thus one can expect that such ITAFs act as chaperones to stabilize the IRES conformation very early in the mRNA maturation process. Although it has not been demonstrated that these ITAFs can bind on the nascent mRNA in a co-transcriptional manner, it is a plausible hypothesis, as co-transcriptional binding has been shown for several hnRNPs during the alternative splicing process (Braunschweig Cell 2013). A sentence has been added line 504-509.

ii) Concerning eIFs, the authors mentioned eIF4A, but nothing is said about eIF3 and other initiation factors. This section is far from being complete.

We agree that the role of eIFs was missing. We have added a paragraph about eIFs, including eIF4A, eIF3, eIF5B and eIF4G lines lines 543-585.

iii) Also, the authors mentioned briefly P-bodies and PCBP1-2. However, the potential translocation of mRNAs and ITAFs to stress granules, a fast cellular response to stress signals, is lacking. iv) The association of ITAFs to the ribosome needs further clarification as well. As it stands, it seems that Reaper is the only protein reported to interact with the ribosome regulating IRES activity.  This is not correct; a few others such as Rack1 and Gemin5 should be included in this point. v) In addition, authors should check the activity of Reaper for IRESs described in flies.

We have developed these points in the text lines 587-611.

vi) A large number of ITAFs are collected in Table 1, yet many of them are not mentioned in the text.

Our choice has been to select different ITAFs in the text (about 30 are mentioned over the list of 50) but not mention all of them to avoid the boring effect of a catalogue and also to limit the length of the text. We prefer to mention the list of ITAFs only in the table.

2.     The first sentences of the introduction should be rewritten. To clarify how cellular IRESs were discovered, it is critical to report the discovery of prototype viral IRES (picornavirus, HCV, and dicistrovirus), and summarize the critical features of these elements. This is relevant for readers outside the field, as cellular IRES are compared later in the text to picornavirus or to viral IRESs several times in the text. In this regard, it will be important to clarify the type of viral IRES compared to, as not all of them behave the same way. 

We agree the description of viral IRESs was to superficial. The text has been developed according to the advice of the reviewer. A specific section has been dedicated to the viral IRESs lines 63-81.

3.     For the sake of clarity, it is important to describe in the introduction that cellular mRNAs carrying IRES elements are translated using a cap-dependent manner. Authors should then discuss how mRNAs switch from cap-dependent to IRES-dependent translation using specific examples of well validated cellular IRESs. The manuscript lacks information about di- or multicistronic cellular mRNAs, and how and when the second cistron could be translated.

A specific section has been dedicated to cellular IRESs lines 83-190. The cap-dependent translation of the cellular IRES-containing mRNAs has been addressed, as well as the switch from cap-dependent to IRES-dependent translation, lines 104-126.  A paragraph about bi- or multicistronic cellular mRNAs has been written, lines 127-139.

4.     It is surprising that the sentence in l. 40 does not cover a reference to HCV IRES, reported much earlier than the retrovirus IRESs. Moreover, prior to the work of Hellen et al., 1993, PTB was reported to be an ITAF by Jang et al., 1989, and Luz and Beck, 1991.

The article by Jang describing PTB binding to EMCV IRES was published in 1990. We agree we missed these articles. These references have been added lines 779 and 783.

5.     In l. 37-39 it should be mentioned that picornaviruses induced the cleavage of factors essential for cap-dependent translation (eIF4G). Therefore, not only these RNAs are uncapped, but they can be translated under cap-dependent translation shutdown. It is surprising that this point is lacking in the review, since this strategy was used by P. Sarnow´s team to discover the first cellular IRESs. 

This has been added lines 53-55.

6.     The sentence in l. 87 needs to be clarified. The work of Weingarten et al (ref 32) claims, but does not prove, the existence of a large number of IRESs in cellular and viral mRNAs. To put things in perspective, the methodology used in this work and the possibility of including false-positives should be discussed. Moreover, the statement that a short sequence could be responsible for local IRES activity is currently an over-interpretation. Given previous controversies regarding certain published data of IRESs present in cellular RNAs it is important to keep close to the results.

This part of the text has been modified according to the reviewer comment, lines 171-173, and the sentence about local IRESs has been removed (as this point is not crucial for this review).

7.     Differences in secondary structures between viral and cellular IRESs should be better discussed (l. 209-211), as for most cellular IRESs RNA structure is not known. In addition, viral IRESs also differ in RNA structure. Therefore, the sentence about the role of PTB in IRES activity regardless of RNA structure conservation is misleading. This paragraph is lacking what is known about the binding site of PTB in cellular and viral IRES.  Information about the binding site of the different ITAFs listed in Table 1 is also lacking.

The difference in secondary structures between viral and cellular IRES has been addressed lines 110-113.

The issue of PTB binding sites is addressed 326-335. As regards, the cellular ITAFs binding sites, we have limited this description to the APAF1 IRES, as our choice in this review is not to go to deeply in the structural aspects, but more on the functional aspects.

8.     The sentence in l. 219-223 regarding ITAFs involved in viral and cellular IRES activity regulation should be modified; it is important to distinguish between binding and mechanism of action. It is surprising that authors first suggest that there are no different mechanisms of action, but then speak about tissue specificity of cellular IRESs. Are there factors conferring tissue specificity? What is the meaning of specific and general ITAFs? May be the mechanism of action is different.

The sentence has been modified lines 341-346. We do not confuse binding and mechanisms of action. All the ITAFs described here have been described for their ability to modulate IRES activity, not only their binding to RNA. We have also modified the sentence about general ITAFs.

9.     The sentence in l. 238 is not complete.

This has been corrected line 362.

10.  Authors should explain what the meaning of eIF4A-binding mechanism is, and its relevance for IRES activity (l. 351).

                It has been explained lines 526-530 with the example of pdcd4.  

Reviewer 2 Report

The authors of this manuscript provide a summary of various ITAFs regulating cellular mRNA translation in response to different stress signals.  While this is considered a timely and potentially important publication, the manuscript lacks adequate classifications of ITAFs as well as in-depth discussions that will likely draw more interests from the broad reader base of this and other scientific journals:

Major concerns:

1.     Line 90:  The authors only briefly mentioned that “dysfunction of IRES-dependent translation has also been related with various pathologies”.  It is important to emphasize that these observations may provide exciting therapeutic opportunities targeting different ITAFs for treatment of cancer and other diseases.  However, there is no follow-up in the rest of the review regarding how this strategy may be further realized to potentially benefit the patients with these diseases.

2.     To help the authors with this aspect of the review, it may be helpful to look at the review article of “Ji et al., IJMS, 2017” for their discussions on this topic, i.e., targeting expression of various p53 ITAFs for the treatment of cancer.  

3.     Line 209: While the cellular IRESes are different from virus IRESes as they do not appear to have a common secondary or tertiary structure like that of virus IRESes, it is intriguing that the same ITAFs may regulate both cellular and virus IRESes or multiple cellular IRESes.  However, the authors should point out that with the new advancements of the cryo-EM technology, etc., the current understanding of the ITAF-IRES interactions may be significantly improved to provide a more definitive answer for such dilemma.

4.     Thus, at minimal, the authors should provide a discussion section to summarize some of the points mentioned by this reviewer as well as any other issues that the authors now may feel necessary to be addressed.

Minor concerns:

1.     Line 138:  It is not necessary to mention that the “selective translation upon eIF-2alpha” is the main focus of this review.  The main focus should be, as stated by the authors in the abstract, regulation of cellular IRESes by ITAFs in response to stress.

2.     There are a few typos and other minor language mistakes in the manuscript.

3.     Line 279:  the subtitle should be revised.

4.     The classification of the different ITAFs and their mechanisms of action are tedious and arbitrary: The authors should combine class II and class III in table 1; additionally, the authors should also combine 4.9 with 4.8; and 4.5 with 4.6, etc., along with corresponding revisions in Fig. 2.

5.     In Fig. 2, 4EPB should be 4EBP.  Also, either use 4E-BP or 4EBP, not both.     

Author Response

We wish to thank the reviewer for his important suggestions which allowed us to improve and complete this manuscript.

The authors of this manuscript provide a summary of various ITAFs regulating cellular mRNA translation in response to different stress signals.  While this is considered a timely and potentially important publication, the manuscript lacks adequate classifications of ITAFs as well as in-depth discussions that will likely draw more interests from the broad reader base of this and other scientific journals:

Major concerns:

1.     Line 90:  The authors only briefly mentioned that “dysfunction of IRES-dependent translation has also been related with various pathologies”.  It is important to emphasize that these observations may provide exciting therapeutic opportunities targeting different ITAFs for treatment of cancer and other diseases.  However, there is no follow-up in the rest of the review regarding how this strategy may be further realized to potentially benefit the patients with these diseases.

A sentence has been added lines 187-190 and it has been addressed in the discussion lines 687-696.

2.     To help the authors with this aspect of the review, it may be helpful to look at the review article of “Ji et al., IJMS, 2017” for their discussions on this topic, i.e., targeting expression of various p53 ITAFs for the treatment of cancer.  

The discussion section has been developed lines 663 to 696.

3.     Line 209: While the cellular IRESes are different from virus IRESes as they do not appear to have a common secondary or tertiary structure like that of virus IRESes, it is intriguing that the same ITAFs may regulate both cellular and virus IRESes or multiple cellular IRESes.  However, the authors should point out that with the new advancements of the cryo-EM technology, etc., the current understanding of the ITAF-IRES interactions may be significantly improved to provide a more definitive answer for such dilemma.

A paragraph about cryo-EM has been added lines 372 to 393.

4.     Thus, at minimal, the authors should provide a discussion section to summarize some of the points mentioned by this reviewer as well as any other issues that the authors now may feel necessary to be addressed.

The discussion section has been developed lines 663 to 696. The example of nutlin-3 has not been mentioned in the discussion as nutlin is an mdm2 inhibitor (and probably acts by inhibiting the ubiquitination of p53), while mdm2 is an activator or the p53 IRES, the nutlin effect does probably not involve the ITAF role of mdm2 (Malbert-Colas 2014).

Minor concerns:

1.     Line 138:  It is not necessary to mention that the “selective translation upon eIF-2alpha” is the main focus of this review.  The main focus should be, as stated by the authors in the abstract, regulation of cellular IRESes by ITAFs in response to stress.

The sentence has been modified lines 227-228: “IRES-dependent translation, the focus of the present review, is the other main mechanism of selective translation upon eIF2a phosphorylation”.

2.     There are a few typos and other minor language mistakes in the manuscript.

We have corrected all the mistakes that we could detect.

3.     Line 279:  the subtitle should be revised.

The subtitle has been changed line 432: “ITAFs use different mechanisms of action to control IRES-dependent translation”.

4.     The classification of the different ITAFs and their mechanisms of action are tedious and arbitrary: The authors should combine class II and class III in table 1; additionally, the authors should also combine 4.9 with 4.8; and 4.5 with 4.6, etc., along with corresponding revisions in Fig. 2.`

This has been changed lines 289-301 and in Table 1. 7.5 has been combined with 7.6. However we have maintained the separate paragraphs 7.9 and 7.8, as 7.8 has been strongly developed in response to reviewer 1 request.

5.     In Fig. 2, 4EPB should be 4EBP.  Also, either use 4E-BP or 4EBP, not both.

                We have chosen “4EBP”. In the revised version, 4EBP does not appear any more in the selected                 examples of Fig. 2.  

Round 2

Reviewer 1 Report

The authors addressed the comments satisfactorily

Author Response

Reviewer 1 did not ask additional correction in his second report. We are happy to have satisfied the requests of the first report.

Reviewer 2 Report

The authors have revised the manuscript according to the comments of both reviewers.  However, this reviewer feels that there are still rooms to further improve this manuscript.  There are some minor mistakes in the manuscript that need to be corrected as well:

1.    Line 185-190 could be revised as:  Another recent study has shown that reduced expression of a couple of positive ITAFs (TCP80 and RHA) of p53 IRES may lead to diminished p53 response following DNA damage in breast cancer cells expressing wild-type p53 (Reference 155).  These different studies of IRES -related pathophysiological functions demonstrate the key role of IRES dependent translation, revealing the coexistence of cap-dependent and independent translation for capped mRNAs containing IRESs. They also provide exciting therapeutic opportunities to specifically regulate the expression of IRES-containing genes (often involved in the control of cell proliferation, cell death, angiogenesis…) at the translational level by targeting different ITAFs.

2.    Line 669-678:  More in-depth discussions are needed to further improve this paragraph (see comments below):

Due to the presence of the cap-structure in eukaryotic mRNAs, the only way to demonstrate the presence of an IRES in eukaryotic mRNAs is to use a bicistronic reporter system.  However, it has been shown that multiple previously reported IRES structures contain cryptic promoters resulting in false-positive findings.  Thus, stringent and carefully controlled experiments, such as those (Northern blot and RT-PCR, etc.) performed in two reports demonstrating the presence of p53 IRES (references 49 and 51), are necessary to demonstrate the presence of a bono fide cellular IRESs.

3.    Line 688, please change “block” to “regulate” (could be either “block” or “stimulate”).

4.    Line 691, please change reference 93 to reference 155.

Author Response

Our answer to the different points is in red italic.

The authors have revised the manuscript according to the comments of both reviewers. 

We are happy that Reviewer 2 has been almost satisfied by our first revision. We have addressed all the minor comments described below.

However, this reviewer feels that there are still rooms to further improve this manuscript.  There are some minor mistakes in the manuscript that need to be corrected as well:

1.     Line 185-190 could be revised as:  Another recent study has shown that reduced expression of a couple of positive ITAFs (TCP80 and RHA) of p53 IRES may lead to diminished p53 response following DNA damage in breast cancer cells expressing wild-type p53 (Reference 155).  These different studies of IRES -related pathophysiological functions demonstrate the key role of IRES dependent translation, revealing the coexistence of cap-dependent and independent translation for capped mRNAs containing IRESs. They also provide exciting therapeutic opportunities to specifically regulate the expression of IRES-containing genes (often involved in the control of cell proliferation, cell death, angiogenesis…) at the translational level by targeting different ITAFs.

The underlined sentence with ref 155 has been included lines 704-707 rather than at line 185. We do not wish to describe cellular ITAFs at this stage of the manuscript, as such description starts only line 278 (section 5). The other corrections have been added line 185 and 188.

2.     Line 669-678:  More in-depth discussions are needed to further improve this paragraph (see comments below): Due to the presence of the cap-structure in eukaryotic mRNAs, the only way to demonstrate the presence of an IRES in eukaryotic mRNAs is to use a bicistronic reporter system.  However, it has been shown that multiple previously reported IRES structures contain cryptic promoters resulting in false-positive findings.  Thus, stringent and carefully controlled experiments, such as those (Northern blot and RT-PCR, etc.) performed in two reports demonstrating the presence of p53 IRES (references 49 and 51), are necessary to demonstrate the presence of a bono fide cellular IRESs.

Indeed we thank the reviewer to have insisted on this point, very important in the IRES field. The suggested revision has been added in a paragraph lines 673-686.

3.    Line 688, please change “block” to “regulate” (could be either “block” or “stimulate”).

  The correction has been made line 701.

4. Line 691, please change reference 93 to reference 155

Indeed we cited a wrong reference. Reference 93 has been changed to reference 155 line 707